# Preparation of 6N,7N High-Purity Gallium by Crystallization: Process Optimization

**DOI:** 10.3390/ma12162549

**Published:** 2019-08-10

**Authors:** Jianfeng Hou, Kefeng Pan, Xihan Tan

**Affiliations:** 1Department of Chemistry and Chemical Engineering, Lvliang University, Lvliang 033001, China; 2School of Metallurgy, Northeastern University, Shenyang 110819, China

**Keywords:** gallium, high-purity, crystallization, process optimization

## Abstract

In this study, radial crystallization purification method under induction was proposed for preparing 6N,7N ultra-high purity gallium crystal seed. The effect of cooling temperature on the morphology of the crystal seed, as well as the cooling water temperature, flow rate, and the addition amount of crystal seed on the crystallization process was explored, and the best purification process parameters were obtained as follows: temperature of the crystal seed preparation, 278 K; temperature and flow rate of the cooling water, 293 K and 40 L·h^−1^, respectively; and number of added crystal seed, six. The effects of temperature and flow rate of the cooling water on the crystallization rate were investigated. The crystallization rate decreased linearly with increasing cooling water temperature, but increased exponentially with increasing cooling water flow. The governing equation of the crystallization rate was experimentally determined, and three purification schemes were proposed. When 4N crude gallium was purified by Scheme I, 6N high-purity gallium was obtained, and 7N high-purity gallium was obtained by Schemes II and III. The purity of high-purity gallium prepared by the three Schemes I, II, and III was 99.999987%, 99.9999958%, and 99.9999958%, respectively.

## 1. Introduction

In the 1970s, compounds comprising gallium and Group IIIA elements were discovered to have excellent semiconductor properties. Since then, gallium (Ga) has been gradually used in the semiconductor industry as a raw material [1]. In recent years, with the continuous development of science and technology and people’s pursuit of low-carbon economy and green energy, the application of Ga has been fully developed and it has become one of the important raw materials in the fields of modern semiconductors (approximately 80% of the total consumption of gallium), solar energy (approximately 10% of the total gallium consumption, magnetic materials (~5% of the total gallium consumption), and catalysts, and has been widely applied in defense, optical fiber communication, aerospace, and other fields [2,3].

At present, the production technology of low-grade gallium (purity ≤ 99.99%) has been perfected day by day [4,5,6]. The statistics from the US Geological Survey (USGS) in 2018 showed that [7] the amount of low-grade primary gallium production around the world in 2017 was ~315 tons, which increased by 15% compared to the amount in 2016 with 274 tons. According to the survey report of USGC in 2015 [8], the global demand for Ga was estimated to increase by 20 times in 2030, whereas with the development of semiconductor devices with higher performance, the demand of high impurity gallium has been increasing, because even very small amounts of impurities such as Cu, Pb, Fe, Mg, Zn, and Cr, which are always present in current large-scale commercial-quality gallium, can degrade or limit the electrical properties [9]. Conventional refining methods such as electrolytic refining [10,11], regional melting [12], vacuum distillation [13], and drawn single crystal method [14] have been applied for the preparation of high-purity gallium, and the electrolytic refining method is the most widely used high-purity gallium production technology in industry at present. However, these traditional methods had many problems such as high energy consumption, lack of environmental friendliness, low production efficiency, and inconvenient automation control. Therefore, developing an advanced purification technology is of great significance to the development of the contemporary semiconductor and the solar industry.

The purification and refining of gallium have been systematically investigated by us [15,16,17]. Based on the traditional crystallization purification method, a radial crystallization purification method was proposed by the seed crystal induced crystallization. The method had the advantages of low energy consumption, simple equipment, convenient operation, and short production cycle. In this study, the crystallization experiment was used to explore the effect of cooling temperature on the morphology of the crystal and the effect of cooling water temperature, flow rate, and the amount of seed crystal added on the crystallization process. The parameters of the purification process were explored in order to optimize the best purification process, determine the crystallization rate control equation, and prepare high-purity (6N and 7N) metal gallium under the process conditions.

## 2. Materials and Methods

### 2.1. Process Design

Figure 1a shows the preparation process of 6N,7N high-purity gallium by radial crystal purification method through seed induction. The main steps and operation methods are as follows:

(1) Cleaning the crystallizer and assembling the purification device

First, the crystallizer was rinsed with ultrapure water (resistivity ≥16 MΩ·cm) to remove the dust on the surface. Then, it was cleaned using an ultrasonic cleaning device containing ultrapure water for 2 h to remove residual contaminants on the surface. The purification device was assembled, as shown in Figure 1b.

(2) Pretreatment of 4N crude gallium

4N crude gallium along with the package bottle was placed on a hot plate, and the heating temperature was set to 335 K. After the gallium was melted, the molten crude gallium was transferred to a polytetrafluoroethylene beaker and mixed with 200 mL of 3 mol/L HCl at 335 K for 2 h. The hydrochloric acid was suctioned out using a plastic pipette, and then 200 mL of 3 mol/L HNO3 was added to the beaker, followed by mixing and stirring for 2 h. The crude gallium was washed with acid, followed by washing with ultrapure water three times. The hydrochloric acid and nitric acid used in the acid treatment were all high purity grades, and ultrapure water was used for preparation of acid solution.

(3) The pretreated molten crude gallium (4N) was transferred into a clean crystallizer.

(4) Circulating cooling water was introduced into the water jacket of the crystallizer.

The cooling water was supplied using a low constant temperature water tank with a built-in circulating water pump. The temperature range was in the range 263–373 K, the temperature control accuracy was ±0.1 K, and the flow rate of the cooling water was controlled using a glass rotor flow meter.

(5) When the temperature of the liquid gallium dropped to the critical point of crystallization, crystal seed was added thereto, and the cooling water was circulated.

The seed crystal was prepared before the start of the purification experiment, using 7N gallium as the raw material. The method used is as follows: a polytetrafluoroethylene beaker containing molten 7N gallium was placed in a low constant temperature water bath to cool and crystallize. Liquid gallium was continuously stirred with a Teflon rod to disperse the crystal nucleus and improve the nucleation rate. The crystallization of liquid gallium was observed in the stirring process. When the grain with the desired size (0.5 cm) was formed, PTFE tweezers were used to pick out the crystal for later use.

(6) When the liquid gallium was crystallized to a preset crystallization ratio, the introduction of cooling water was stopped, and the residual liquid gallium was discharged out of the crystallizer.

(7) A three-way switch was switched, and the circulating hot water was introduced into the cooling/melting zone of the crystallizer. After the solid gallium completely melted, the three-way switch was switched, and the circulating cooling water was reintroduced into the cooling/melting zone;

The hot water was supplied using a constant temperature water tank with a built-in circulating water pump. The temperature range was 278–373 K, and the temperature control accuracy was ±1 K.

(8) The steps (4)–(7) (as shown in Figure 1c) were repeated to a predetermined number of crystallizations, and after the completion of purification, the product quality was detected.

### 2.2. Detection Method

In the experiment, the impurity contents of the 4N raw gallium material and the purified 6N,7N high-purity gallium were detected by high-resolution glow discharge mass spectrometry (Evans Materials Technology (Shanghai) Co., China, HR-GDMS), and the purity of the product was calculated by the difference method. Argon was used as the discharge gas for detection. The mass spectrometry parameters are as follows: discharge current, 1.9 mA; discharge voltage, 1 kV; beam current of gallium ion, 1 × 10^−6^ mA; insulating layer, aluminum; and resolution ≥ 3600. Before data collection, the ion source of the HR-GDMS was cooled to the temperature of liquid nitrogen (90 K) to reduce the ion interference in the background gas. Then, the surface of the tested sample (0.2 × 2 mm^2^) was pre-sputtered for 5 min at liquid nitrogen temperature to remove the contaminants from the sample surface. The pre-sputtering conditions were kept constant, and data collection was started. During the data collection process, the integration time was set as 80 ms.

## 3. Results and Discussion

### 3.1. Effect of Cooling Temperature on Seed Morphology

The appearance morphology of the seed crystal prepared at the cooling temperature in the range 265–295 K is shown in Figure 2, indicating that at 265 K, the solidified structure comprised many fine crystal grains, and the grains were interspersed with a large amount of liquid gallium. When the solidified structure was removed, a large amount of liquid gallium was attached to the surface, resulting in an extremely irregular shape of crystal seed, because at 265 K, the growth rate of crystal nucleus increased after nucleation due to the large degree of supercooling, leading to the emergence of a large number of dendrites. The rapid growth of dendrites not only mixed with a lot of liquid phase inside the solidified structure, but also caused a lot of hollow surface of the solidified structure. When the preparation temperature was 273 K, the solidified structure exhibited the geometric polyhedron shape characteristics, indicating that with decreasing subcooling degree, the growth rate of crystal nucleus decreased, and its growth mode transited from dendrite growth to lamellar growth. When the preparation temperature was 278 K, the solidified structure showed a regular hexahedral shape, suggesting that with increasing temperature, the subcooling degree of the growth front further reduced after the formation of crystal nucleus, and the growth mode changed into vertical layered growth. When the preparation temperature was 295 K, the supercooling degree of the solid–liquid interface further decreased after the crystal nucleus formed, hindering the release of latent heat from crystallization. At this time, in order to release the latent heat of crystallization at a faster rate, the growth direction of crystal nucleus changed to side growth, distorting its geometric shape. By comparing the morphological characteristics of crystal seed prepared at four temperatures, the optimal preparation temperature of the crystal seed was finally determined to be 278 K.

### 3.2. Effect of Process Parameters on the Crystallization Process

#### 3.2.1. Effect of Temperature of Cooling Water on Crystallization Process

When the flow rate of cooling water was 40 L·h^−1^ and the temperature was in the range 288–298 K, 2.9774 kg crude gallium obtained by pickling pretreatment was cooled to the critical point of crystallization, followed by adding crystal seeds for 15 min. The corresponding crystal growth morphology is shown in Figure 3.

Figure 3 shows that when the temperature of the cooling water were 288 and 290 K, the crystal growth mode of liquid gallium was mainly dendrite growth after adding the crystal seed, and the crystal branches bridged with each other, exhibiting liquid gallium trapped inside the crystal. This was because at lower cooling water temperature, the temperature gradient inside the liquid gallium was higher, and the growth rate of the crystal was faster after adding the seed crystal. Although the positive temperature gradient was formed at this time, the temperature at the front of the solid–liquid interface was higher in the radial direction of the crystallizer, hindering the release of latent heat of crystallization in this direction, and thus decreasing the crystal growth in this direction. However, in order to facilitate the release of latent heat of crystallization, the growth orientation of the crystal changed and grew rapidly in the form of dendrites, eventually bridging the crystal branches and entrapment of the liquid phase. The entrained liquid phase impurities cannot be removed because of the growth mode of crystal, thus affecting the purification process. When the temperature of cooling water was 293 K, the liquid gallium grew into a single crystal after adding the seed crystals. 

In order to further analyze the growth law of the liquid gallium during crystallization, the crystal morphology at different times after adding the crystal seeds was investigated by the dynamic timing observation method, when the cooling water flow was 40 L·h^−1^ and the temperature was 293 K. The result is shown in Figure 4.

Figure 4 shows that after the addition of the crystal seed, the gallium crystal block gradually grew with increasing crystallization time, and the crystal growth mode of liquid gallium exhibited typical layer-by-layer push growth after adding the seed crystals, indicating that the temperature gradient environment formed by the cooling water at 293 K could release the latent heat of crystallization generated during the crystal growth to the front of the solid–liquid interface, and transferred and released it outwardly along the direction of temperature gradient. This kind of layer-by-layer crystal growth mode was beneficial to the enrichment of impurity elements from the solid–liquid interface to the liquid phase, thereby affording the solid Ga metal with higher purity. The supercooling degree of the growth tip was the largest when the crystal grew, and the liquid gallium atoms at the solid–liquid interface preferentially attached to the growth tip, and the heat was transferred outward from the crystallized solid gallium in the direction of positive temperature gradient in the crystallizer. Therefore, in the crystallization process, crystal growth was always in the form of a pyramind-shaped step-by-layer advancement. According to the crystal growth kinetics and thermodynamics, the layer-by-layer growth method was conducive to increase the surface area of crystals, facilitating the release of latent heat of crystallization and ensuring the continuous and steady growth of crystals during the crystallization process. Moreover, according to the separation and coagulation theory of impurities in the crystallization process, the layer-by-layer growth mode was favorable to the enrichment of impurity elements from the solid–liquid boundary to the liquid state and can avoid the inclusion impurities of liquid-phase envelopment caused by irregular crystal growth direction.

Figure 4 shows that with increasing crystallization time, the pyramind tip of crystal became more and more obvious, and the layered step of crystal growth was also more and more obvious, attributed to the fact that as the crystallization continued, impurity elements constantly accumulated in the liquid phase, and the impurity content at the solid–liquid interface increased, which enhanced the probability of impurity elements attaching to the crystal growth tip. Owing to the difference in the atomic radius and electronegativity between Ga and impurity elements, the impurity atoms attached to the growth tip entered into the Ga lattice or lattice gap, causing the growth defect of Ga crystal [18,19,20]. This indicated that the removal of impurity elements decreased with the progress of crystallization and was consistent with the literature data [16].

#### 3.2.2. Effect of Cooling Water Flow on Crystallization Process

In a previous study, the effect of cooling water flow on the crystallization process was primarily investigated [16]. The results revealed that when the cooling water flow rate was 30 L·h^−1^, the growth rate of gallium crystal near the outlet of the crystallizer was slightly lower than that in other regions. When the cooling water flow rate was 50 L·h^−1^, the growth rate of the gallium crystal in the lower part of the crystallizer was slightly larger than that of the upper part, and the growth rate near the crystallizer inlet was the largest. When the cooling water flow rate was 40 L·h^−1^, the growth rate of gallium crystals in all the regions of the crystallizer was basically the same, and too fast or too slow local growth phenomenon was not observed. In order to further explore the effect of this process parameter on the crystallization process, the crystal morphology of liquid gallium at different cooling water flow rates was observed, and the results are shown in Figure 5.

Figure 5 shows that at a cooling water flow rate of 40 L·h^−1^, the crystal morphology of gallium exhibited a distinct “shell pattern” with the uniform grain spacing. This indicated that under that flow rate, the gallium crystal grew in the layer-by-layer manner and was favorable to remove the impurity. At a cooling water flow rate of 30 L·h^−1^, the crystal growth rate at the outlet side of the crystallizer was slightly slower than that in other areas, and its crystal morphology was the same as that at a cooling water flow of 40 L·h^−1^, also displaying a distinct “shell pattern”. This suggested that under this flow condition, the gallium crystals also grew in the layer-by-layer manner, which was favorable for the removal of impurity; however, the crystal growth rate here was slower than its surrounding region, and thus the possibility of enveloping the liquid phase with the progress of crystallization at this point cannot be ruled out. However, at a cooling water flow rate of 50 L·h^−1^, due to the higher heat transfer efficiency at the bottom of the outlet side of the crystallizer, the driving force of gallium crystal growth was larger and the crystal growth rate was faster, changing the crystal morphology and presence of a large number of irregular growth steps. It can be deduced that the crystals at the site were not completely in the way of layer-by-layer promotion growth, and the crystal growth process may be accompanied by dendrite or peritectic formation, leading to envelope liquid phase, entrap impurities, and reduce purification effect in solid gallium.

#### 3.2.3. Effect of Seed Number on Crystallization Process

When the cooling water flow rate was 40 L·h^−1^ and the temperature was 293 K, the liquid gallium cooled to the critical point of crystallization and 3, 4, 5, and 6 crystal seeds were added. When the crystallization reached a certain proportion, its morphological image is shown in Figure 6, indicating that the number of seed added determined the shape of the uncrystallized region. When three crystal seeds were added, the uncrystallized region exhibited a triangle shape. When four crystal seeds were added, the uncrystallized region exhibited a quadrilateral shape. However, when the number of added crystal seed was 3 or 4, the shape and size of the uncrystallized area were not consistent, showing a funnel shape with a large top and a small bottom. This easily led to the intersection of the crystal growth at the bottom of the crystallizer with the continuous progress of crystallization, which caused the envelopment of the liquid phase and entrapment of impurities, thus affecting the purification effect. When five seed crystals were added, the uncrystallized region presented a pentagon shape, and the problem of shape with a large top and small bottom in the uncrystallized region improved. In the case of adding six seed crystals, the uncrystallized region displayed a hexagonal shape with a regular shape and uniform size and was most beneficial to control the overall direction of the crystal during the purification of crude gallium. Therefore, the optimal number of seed addition was determined to be six, when the 4N crude gallium was purified using a self-made crystallizer.

### 3.3. Effect of Process Parameters on Crystallization Rate

In the actual crystallization solidification process of liquid gallium, the crystallization rate (i.e., the crystal growth rate of gallium after adding crystal seed) depended on the supercooling degree of the solid–liquid interface. The supercooling degree of solid–liquid interface was a function of temperature and cooling water flow keeping other process conditions constant. In the experiment, the relationships between the crystallization rate and cooling water temperature as well as the flow were measured by the control variable method, and the empirical control formula of the crystallization rate was obtained by analyzing the experimental data. In order to reduce the experimental error, improve the accuracy of empirical control formula and its adaptability to the actual production process, each group of measurement experiment was repeated four times, and the average value was taken. The crystallization rate measured in the experiment was the average rate during the complete solidification process of liquid gallium after adding the crystal seed, and the calculation formula is as follows:(1)$$v=\frac{m}{t}$$
where *v* is the average rate, kg/h; *m* is the total mass of liquid gallium, kg; *t* is the time required for the complete solidification of liquid gallium, hour (h).

The effect of the temperature and flow rate of the cooling water on the crystallization rate determined by the test is shown in Figure 7. Figure 7a shows that with increasing cooling water temperature, the crystallization rate gradually decreased, and an obvious linear relationship was observed between the two. The empirical control formula of the cooling water temperature on the crystallization rate was obtained by Origin software fitting.
*v*(*T*) = −0.09*T* + 27(2)
where *T* is the temperature of the cooling water, K; and the linear correlation coefficient of the data fit was R^2^ = 0.997.

Figure 7b shows that with increasing flow rate of cooling water, the crystallization rate increases, and a significant exponential function relationship was observed between the two. The empirical control formula of the cooling water flow to the crystallization rate was obtained by Origin software fitting.
(3)$$v(Q)=-96.73e^{\frac{-Q}{4.94}}+0.66$$
where *Q* is the flow rate of the cooling water, L/h; and the standard deviation of the data fit was R^2^ = 0.997.

### 3.4. Analysis of Purification Results

Based on the above studies, the optimum technological parameters for the crystal purification of 4N raw material gallium were determined as follows: the temperature of seed preparation, 278 K; cooling water temperature, 293 K, cooling water flow, 40 L·h^−1^, and the number of seed crystals added was six. Combined with our previous research results [16], three purification schemes were determined under the optimized process parameters, as listed in Table 1.

The impurity contents in high-purity gallium prepared by the three purification schemes were tested and compared to the raw gallium material, and the removal rate of impurities was calculated. The results are shown in Table 2.

Table 2 shows that for Scheme I, after the purification, the mass fraction of Al impurities contained in the raw materials reduced below the detection limit of HR-GDMS, and the other six main impurities were also well removed. The removal rates were: Fe-87.1%, Pb-95.9%, Zn-89.9%, Mg-97.9%, Cu-98.8%, and Cr-93.3%, and the mass fraction of the main gallium metal in the refined high-purity Ga calculated by the difference method was 99.999987%. For Scheme II, the removal rates of the six main impurities were Fe-93.8%, Pb-98.8%, Zn-95.6%, Mg-99.6%, Cu-99.8%, and Cr-97.6%, and the mass fraction of the main Ga metal was 99.9999958%. For Scheme III, the removal rates of the six major impurities further increased, and the removal rates of Mg and Cu exceeded 99.9%. In contrast, although the removal rate of Fe was the lowest, it also reached 97%. The mass fraction of main Ga metal was 99.9999958%.

## 4. Conclusions

In conclusion, the impurity removal of Ga was investigated in detail, and the radial crystallization purification method under the seed crystal induction is proposed. The effect of cooling temperature on the morphology of the crystal as well as the cooling water temperature, flow rate, and the number of crystal seeds added on the crystallization process were investigated. The optimum purification process was obtained; the control equation of the crystallization rate was determined; and the high-purity (6N and 7N) gallium was prepared under the technological conditions. The main conclusions of this study are as follows:(1)The optimum process parameters for the crystallization purification of 4N raw material gallium are as follows: temperature of the seed preparation, 278 K; cooling water temperature, 293 K; cooling water flow, 40 L·h^−1^; the number of seed crystals added six 6;(2)The crystallization rate decreased linearly with increasing cooling water temperature and increased exponentially with increasing cooling water flow. The control formulas of the cooling water temperature *T* and flow *Q* on the crystallization rate *v* are, *v*(*T*) = −0.09*T* + 27 and $v(Q)=-96.73e^{\frac{-Q}{4.94}}+0.66$, respectively;(3)The three proposed purification schemes effectively removed the impurity elements. When using Scheme I to purify the 4N crude gallium, high-purity gallium with a purity of 6N was obtained. When adopting Schemes II and III, 7N high-purity gallium was obtained. The purities of the high-purity gallium prepared by Schemes I, II, and III were 99.999987%, 99.9999958%, and 99.9999958%, respectively.

The seed crystal induced radial crystallization purification method proposed in the study has the advantages of simple operation, convenient process flow, low energy consumption, environmentally friendly, and easy to realize automatic control of the purification process, providing a new idea for the large-scale industrial production of ultra-high purity gallium.

## Figures and Tables

**Figure 1 materials-12-02549-f001:**
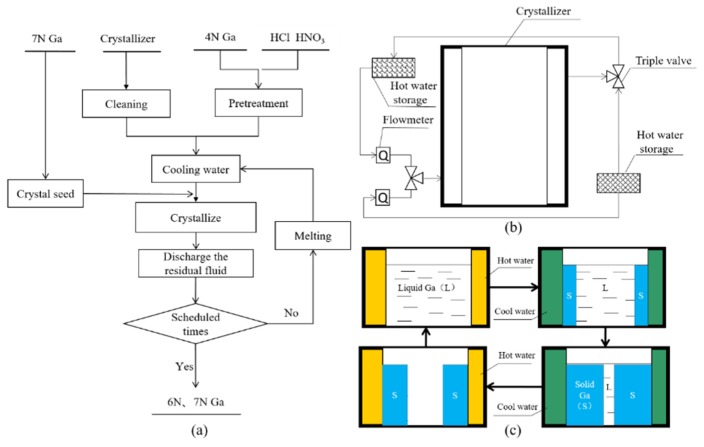
Schematic diagram of the purification process: (**a**) Process flow for purification, (**b**) assembly of the purification device, and (**c**) repeated crystallization process.

**Figure 2 materials-12-02549-f002:**
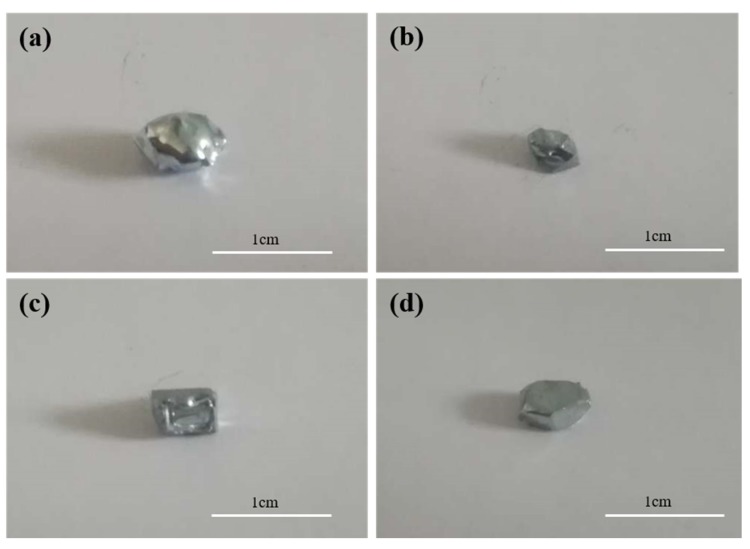
Morphology of the seed crystals prepared at different cooling temperatures ((**a**) 265 K, (**b**) 273 K, (**c**) 278 K, and (**d**) 295 K).

**Figure 3 materials-12-02549-f003:**
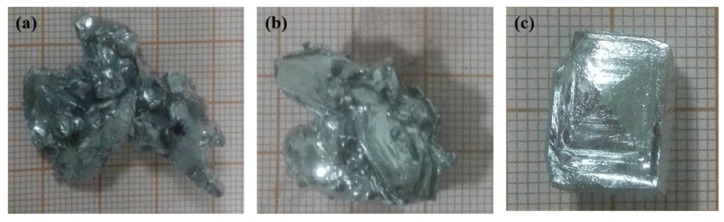
Crystalline morphology of liquid gallium after adding crystal seeds for 15 min at different temperatures of cooling water ((**a**) 288 K, (**b**) 290 K, and (**c**) 293 K).

**Figure 4 materials-12-02549-f004:**
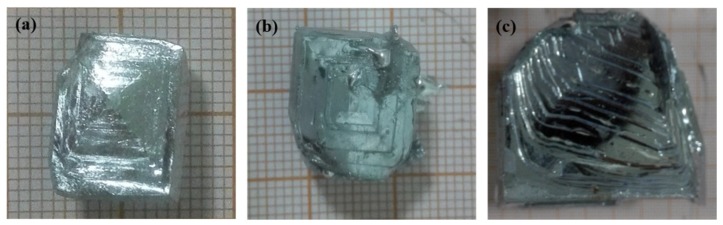
Morphological images of crystal blocks at different times after adding crystal seed ((**a**) 15 min, (**b**) 30 min, and (**c**) 60 min).

**Figure 5 materials-12-02549-f005:**
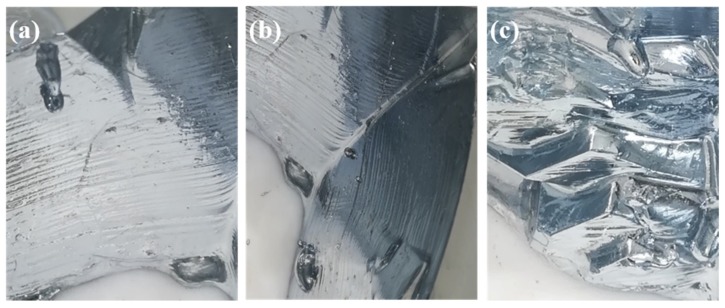
(**a**) At 30 L·h^−1^, near the crystallizer outlet, (**b**) at 40 L·h^−1^, near the crystallizer inlet, and (**c**) at 50 L·h^−1^, detail morphology of gallium crystal near the crystallizer inlet.

**Figure 6 materials-12-02549-f006:**
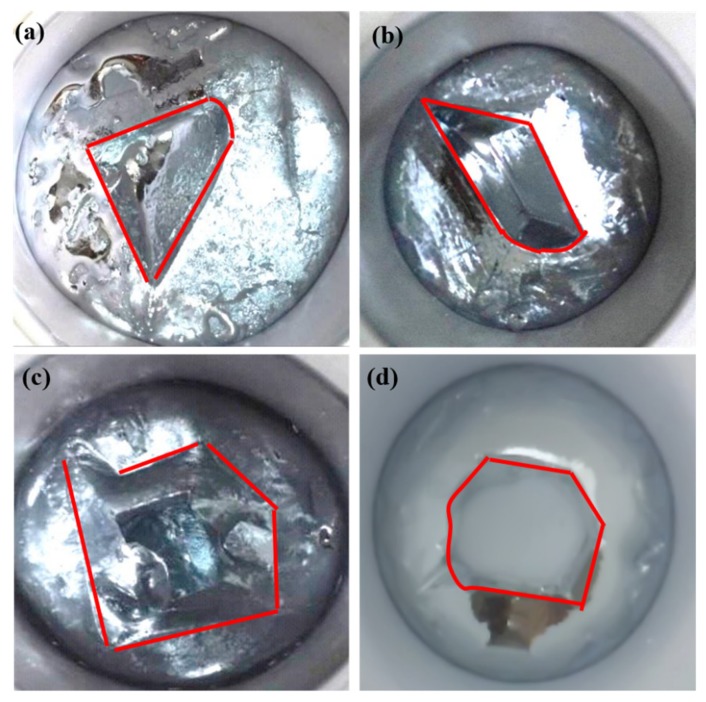
Photographs of crystal morphology when adding different numbers of seeds ((**a**) 3, (**b**) 4, (**c**) 5, and (**d**) 6).

**Figure 7 materials-12-02549-f007:**
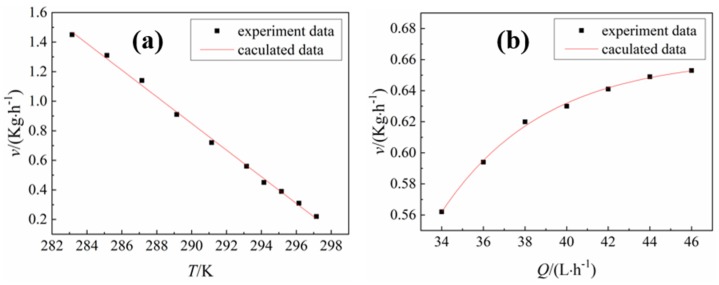
(**a**) Effect of cooling water temperature on the crystallization rate at a flow rate of 40 L·h^−1^; (**b**) effect of cooling water flow rate on the crystallization rate at 293 K.

**Table 1 materials-12-02549-t001:** Scheme of preparing 6N,7N high-purity gallium by the crystal method.

Scheme Number	Recrystallization Number	Crystal Proportion %			
		First Time	Second Time	Third Time	Forth Time
I	2	70	70	/	/
II	3	70	70	85	/
III	4	70	70	85	85

**Table 2 materials-12-02549-t002:** Detection of impurity contents in raw gallium material and high purity purified gallium.

Element	Mass Fraction (×10^−6^)				Removal Rate (%)		
	Raw Material	I	II	III	I	II	III
Mg	76	1.59	0.34	0.07	97.9	99.6	99.9
Al	1	<0.1	<0.1	<0.1	≈100	≈100	≈100
Cr	40	2.66	0.94	0.34	93.3	97.6	99.2
Fe	15	1.94	0.93	0.45	87.1	93.8	97.0
Cu	107	1.32	0.22	0.04	98.8	99.8	>99.9
Zn	24	2.42	1.05	0.45	89.9	95.6	98.2
Pb	56	2.32	0.68	0.20	95.9	98.8	99.6

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
