# Peer review of "Preparation of 6N,7N High-Purity Gallium by Crystallization: Process Optimization"

_materials, 2019, doi:10.3390/ma12162549_

Round 1

Reviewer 1 Report

Detailed Comments and Suggestions can be found in the attached file

Author Response

Dear Reviewer:

Thank you for your comments concerning our manuscript entitled “Preparation of 6N, 7N High-Purity Gallium by Crystallization: Process Optimization” (ID: materials-563343). Those comments are all valuable and very helpful for revising and improving our paper, as well as the important guiding significance to our researches. We have studied comment carefully and have made correction which we hope meet with approval. Revised portion are marked using the "Track change" function in Microsoft Word in the paper. The main corrections in the paper and the responds to the reviewer’s comments are as flowing:

1. Comment: Line 165 ( referred to Figure 3c). Actually one does observe (in Fig. 3c) a regular single crystal: nevertheless, the morphological quality of its faces is poor. This implies that there is not a clear sign of the mechanisms ruling the growth of the faces. Nobody would be able to say their character (flat, stepped or kinked) and then the presence of 2D-nucleation or spiral growth (layer by layer growth).
On the contrary, growth layers (macro-steps) can be found in Fig. 4b and layers bunching in Fig. 4c.
Moreover, the term “cone tip”, used at line 185, should be changed in “pyramid –tip” since the growth layer are polygonised

Response: (1)We revised descriptions of figure 3 and figure 4. 

"When the temperature of cooling water was 293 K, the liquid gallium grew into a single crystal after adding the seed crystals. "

"Figure 4 shows that after the addition of the crystal seed, the gallium crystal block gradually grew with increasing crystallization time, and the crystal growth mode of liquid gallium exhibited typical layer-by-layer push growth after adding the seed crystals, indicating that the temperature gradient environment formed by the cooling water at 293 K could release the latent heat of crystallization generated during the crystal growth to the front of the solid–liquid interface, and transferred and released it outwardly along the direction of temperature gradient. This kind of layer-by-layer crystal growth mode was beneficial to the enrichment of impurity elements from the solid–liquid interface to the liquid phase, thereby affording the solid Ga metal with higher purity. "

(2) We changed "cone tip" to "pyramid tip" in the text.

"Figure 4 shows that with increasing crystallization time, the pyramind tip of crystal became more and more obvious……"

2. Comment: Line 263. The average rate is indicated as v, I suppose. Further, what’s the meaning of h (hours ?). Moreover, the physical dimensions of the quantities v, Q should be indicated also in the text.

Response: (1) We revised descriptions of Eq.(1)

"where v is the average rate, kg/h; m is the total mass of liquid gallium, kg; t is the time required for the complete solidification of liquid gallium, hour (h)."

(2) The physical dimensions of the quantities v, T ,Q have been indicated in the text.

"Where T is the temperature of the cooling water, K; and the linear correlation coefficient of the data fit was R2 = 0.997."

"Where Q is the flow rate of the cooling water, L/h; and the standard deviation of the data fit was R2 = 0.997."

We tried our best to improve the manuscript and made some changes in the manuscript. These changes will not influence the content and framework of the paper. And here we listed the changes. We appreciate for Editors/Reviewers’ warm work earnestly, and hope that the correction will meet with approval.

Once again, thank you very much for your comments and suggestions.

Reviewer 2 Report

Dear Authors, this is an interesting article.

The proposed purification process is good. Validation of project assumptions has been confirmed by the obtained results.

Figure 2. Morphology of the seed crystals prepared at different cooling temperatures - No scale on the pictures a-d. Please correct this.

The described process has potential in industrial application.

Yours faithfully
reviewer

Author Response

Dear Reviewer:

Thank you for your comments concerning our manuscript entitled “Preparation of 6N, 7N High-Purity Gallium by Crystallization: Process Optimization” (ID: materials-563343). your comments are all valuable and very helpful for revising and improving our paper, as well as the important guiding significance to our researches. We have studied comment carefully and have made correction which we hope meet with approval. Revised portion are marked using the "Track change" function in Microsoft Word in the paper. The main corrections in the paper and the responds to the reviewer’s comments are as flowing:

Comment:Figure 2. Morphology of the seed crystals prepared at different cooling temperatures - No scale on the pictures a-d. Please correct this.

Response:A new figure with scale was used to replace the original picture.

Once again, thank you very much for your comments and suggestions.